# Structure-function relationship of ASH1L and histone H3K36 and H3K4 methylation

Kendra R. Vann [1,10], Rajal Sharma[2,10], Chih-Chao Hsu[3], Maeva Devoucoux [4], Adam H. Tencer[1], Lei Zeng [2], Kevin Lin[3], Li Zhu[5], Qin Li[6], Catherine Lachance[4], Ruben Rosas Ospina[1], Qiong Tong[1], Ka Lung Cheung[2], Shuai Yang[2], Soumi Biswas[1], Hongwen Xuan[7], Jovylyn Gatchalian[1], Lorena Alamillo [1], Jianlong Wang [8], Suk Min Jang[4], Brianna J. Klein[1], Yue Lu[3], Patricia Ernst[1], Brian D. Strahl [9], Scott B. Rothbart [7,9], Martin J. Walsh [2], Michael L. Cleary [5], Jacques Côté [4], Xiaobing Shi [3,7], Ming-Ming Zhou [2] ✉ & Tatiana G. Kutateladze [1] ✉

The histone H3K36-specific methyltransferase ASH1L plays a critical role in development and is frequently dysregulated in human diseases, particularly cancer. Here, we report on the biological functions of the C-terminal region of ASH1L encompassing a bromodomain (ASH1L$_{BD}$), a plant homeodomain (ASH1L$_{PHD}$) finger, and a bromo-adjacent homology (ASH1L$_{BAH}$) domain, structurally characterize these domains, describe their mechanisms of action, and explore functional crosstalk between them. We find that ASH1L$_{PHD}$ recognizes H3K4me2/3, whereas the neighboring ASH1L$_{BD}$ and ASH1L$_{BAH}$ have DNA binding activities. The DNA binding function of ASH1L$_{BAH}$ is a driving force for the association of ASH1L with the linker DNA in the nucleosome, and the large interface with ASH1L$_{PHD}$ stabilizes the ASH1L$_{BAH}$ fold, merging two domains into a single module. We show that ASH1L is involved in embryonic stem cell differentiation and co-localizes with H3K4me3 but not with H3K36me2 at transcription start sites of target genes and genome wide, and that the interaction of ASH1L$_{PHD}$ with H3K4me3 is inhibitory to the H3K36me2-specific catalytic activity of ASH1L. Our findings shed light on the mechanistic details by which the C-terminal domains of ASH1L associate with chromatin and regulate the enzymatic function of ASH1L.

Amplification and genetic alterations of ASH1L (absent, small or homeotic discs 1-like) have been linked to a wide array of human diseases, including cancer and autoimmune, developmental and neurodegenerative disorders[1–4]. ASH1L was originally identified as a member of the Trithorax protein family that activates transcription of developmental genes and counteracts Polycomb (Plc) group-mediated gene silencing[5]. ASH1L is conserved from *Drosophila* to humans, and a number of studies have demonstrated that Ash1l is involved in the de-repression of repressed genes, activation of transcription, and shielding of gene bodies from the spreading of the repressive

[1]Department of Pharmacology, University of Colorado School of Medicine, Aurora, CO 80045, USA. [2]Department of Pharmacological Sciences, Icahn School of Medicine at Mount Sinai, New York, NY 10029, USA. [3]Department of Epigenetics and Molecular Carcinogenesis, The University of Texas MD Anderson Cancer Center, Houston, TX 77030, USA. [4]St-Patrick Research Group in Basic Oncology, Oncology Division of CHU de Québec-Université Laval Research, Laval University Cancer Research Center, Quebec City, Québec G1R 3S3, Canada. [5]Department of Pathology, Stanford University School of Medicine, Stanford, CA, USA. [6]Department of Genetics, University of Pennsylvania, Philadelphia, PA, USA. [7]Department of Epigenetics, Van Andel Research Institute, Grand Rapids, MI 49503, USA. [8]Department of Medicine, Columbia Center for Human Development, Columbia University Irving Medical Center, New York, NY 10032, USA. [9]Department of Biochemistry & Biophysics, The University of North Carolina School of Medicine, Chapel Hill, NC 27599, USA. [10]These authors contributed equally: Kendra R. Vann, Rajal Sharma. ✉e-mail: ming-ming.zhou@mssm.edu; tatiana.kutateladze@cuanschutz.edu

machinery[2,6,7]. However, Ash1l has also been shown to maintain repression of poorly transcribed genes, both negatively and positively modulate accumulation of the repressive mark H3K27me3[8] and directly mediate a long-term gene repression[9]. A new model shows that Ash1l may prevent erroneous repression by Plc complexes and there is no correlation between Ash1l functioning and the resistance to Plc repression[10,11].

Human ASH1L plays a key role in establishing transcriptional programs during development and stem cell maintenance and self-renewal. To maintain these programs, ASH1L cooperates with the histone H3K4-specific methyltransferase MLL1[4,12]. Genetic manipulations of *ASH1L* lead to inhibition of hematopoietic stem cells differentiation and reduce expression of developmental genes[5,9,13]. ASH1L itself functions as a specific methyltransferase: it methylates lysine 36 of histone H3 producing mono- and di-methylated epigenetic marks H3K36me1/2[14,15]. The catalytic activity of ASH1L can be stimulated upon formation of the complex with MRG15[16–18], as MRG15 allosterically alleviates the blockade of the ASH1L catalytic site imposed by its autoinhibitory loop[16–18]. Due to its large size (over 300 kDa), biochemical or functional characterization of ASH1L remains particularly challenging. There is no information available about the N-terminal 2,000 amino acids of ASH1L, though several domains have been identified in the C-terminal part, which contains the catalytic SET (Su(var), E(z), Trithorax) domain, followed by a bromodomain (BD), a plant homeodomain (PHD) finger and a bromo-adjacent homology (BAH) domain. Whereas the SET domain has been well characterized[15–18], much less is known about functions of other domains.

Here, we investigate the role of Ash1l in differentiation and gene regulation, characterize ASH1L association with chromatin, and report the structure-function relationship of the C-terminal domains of ASH1L. Our cell, structural, biochemical and enzymatic studies reveal molecular mechanisms for the interactions of BD, PHD and BAH domains of ASH1L and suggest interplay between the epigenetic marks H3K4me3 and H3K36me2 that act, respectively, as a ligand and a product of the catalytic activity of ASH1L.

## Results

### Ash1l expression increases during ES cell differentiation

To assess whether Ash1l functions in a lineage specific manner, we analyzed the RNA-seq data collected from mouse ES cells that were differentiated to a variety of lineages, including mesendoderm, neural progenitor cells (NPCs), neural stem cells (NSCs), retinoic acid (RA) differentiation, and –LIF differentiation (all lineages)[19]. We compared the time-dependent transcriptional expression of *Ash1l* and a selected group of genes that encode key epigenetic mediators, including the Set2 family of H3K36-specific methyltransferases, the Mll family of H3K4-specific methyltransferases, the BET (bromodomain and extra-terminal domain) family of transcriptional activators, and the repressive Polycomb (Plc) family (Fig. 1a). We found that *Ash1l* expression increased during differentiation and was upregulated 1.96 fold (FDR < 0.01) after 4 days of RA differentiation (Fig. 1a, b and Supplementary Fig. 1). Another member of the Set2 family, *Nsd1*, as well as all genes of the Mll family and several Plc genes, such as *Ezh1*, *Ring1* and *Cbx7/8*, were noticeably up-regulated in response to RA differentiation, whereas *Setd2*, the BET family genes *BRD2/3/4*, and Plc *Ezh2* were down-regulated (Fig. 1a). The upregulation of *Ash1l* was confirmed by western blot of the Ash1l protein expression level, which was highest in MEFs or after mouse ES cells were treated with retinoic acid for 6 days. Consistent with the RNA-seq and protein expression data, we observed that Ash1l underwent a dramatic and time-dependent increase in transcriptional expression in embryoid bodies (EBs) differentiated from mouse ES cells (Fig. 1c) and in RA-induced differentiation (Fig. 1d).

### Depletion of Ash1l alters ES cell differentiation

To determine the role of Ash1l in mouse ES cell differentiation, we depleted *Ash1l* by gene knockout or knockdown and monitored expression levels of key differentiation and developmental genes (Supplementary Fig. 2). The CRISPR/Cas9 edited B7 clone showed altered expression of *HoxA9*, *Runx1*, *Neurog2*, and *Olig1* when mouse ES cells were treated with retinoic acid for two days (Fig. 1e). Similarly, when Ash1l knockdown (shAsh1l) cells were allowed to differentiate by formation of embryoid bodies, the timing of transcriptional expression was altered for several differentiation markers, including *HoxA9*, *Pax6*, *Gata4* and *Sox7* (Fig. 1f and Supplementary Fig. 3), consistent with previous reports[4,20]. Furthermore, ChIP-qPCR analysis revealed that upon RA-induced ESC differentiation Ash1l is recruited to promoter and enhancer sites of its target gene *Meis2*, known to undergo transcriptional up-regulation during neuronal development[21] (Fig. 1g). Collectively, these data suggest that Ash1l is essential in regulation of expression of differentiation markers in ES cells.

The importance of Ash1L in transcriptional regulation was substantiated by chromatin immunoprecipitation coupled with deep sequencing (ChIP-seq) analysis of Ash1l in mouse ES cells treated with retinoic acid for 48 h. We found that 64% of Ash1l is present at gene promoters, 27% in gene bodies, and 9% at intergenic sites (Fig. 2a). Gene ontology (GO) analysis confirmed that in ES cells Ash1l is associated with genes involved in transcription and other DNA-templated processes (Supplementary Fig. 4a). The genome-wide occupancy of Ash1l centered around the transcription start sites (TSS) and correlated well with the distribution of the trimethyllysine mark H3K4me3 (Fig. 2b–d and Supplementary Fig. 4b), however the same sites were essentially depleted of H3K36me2 (Fig. 2c, d). The negative correlation between Ash1l/H3K4me3 levels and H3K36me2 levels at TSS was also observed at promoters of individual genes, such as Mef2d (Fig. 2e). Overall, these results suggest that Ash1l may not be fully catalytically active at TSS, which are enriched in H3K4me3.

### ASH1L co-localizes with H3K4me3 but not with H3K36me2

To test if the negative correlation with H3K36me2 is conserved, we analyzed previously reported ChIP-seq datasets from the human MV4-11 cell line and assessed localization of ASH1L, H3K36me2 and H3K4me3[4] (Fig. 2f–h). Much like in the mouse ES cells, peak-centered evaluation showed that ASH1L occupancy is highest around the TSS in the human MV4-11 cells (Fig. 2f). ASH1L co-localized with H3K4me3 but H3K36me2 was excluded from the ASH1L sites at TSS, instead co-localizing with ASH1L at gene bodies downstream of TSS. As anticipated, ASH1L bound sites positively correlated with occupancies of MLL1, a methyltransferase that generates H3K4me3 and MLL1's binding partner LEDGF. H3K36me3 levels were essentially undetected at TSS and peaked at the transcription termination sites (TTS). A subtle negative correlation between H3K36me2 and H3K4me3 signals was also observed in Pearson correlation analysis of genome-wide signals (Fig. 2g).

### ASH1L$_{PHD}$ recognizes H3K4me2/3

The H3K36me2/1-specific catalytic SET domain of ASH1L (ASH1L$_{SET}$) is followed by a bromodomain (ASH1L$_{BD}$), a PHD finger (ASH1L$_{PHD}$) and a BAH domain (ASH1L$_{BAH}$) that may act as chromatin binding modules (Fig. 3a). PHD fingers are known to recognize histone sequences[22–26], and we confirmed that ASH1L$_{PHD}$ is a histone reader using NMR, fluorescence spectroscopy, and microscale thermophoresis (MST) (Fig. 3b–e and Supplementary Fig. 5). Titration of H3K4me3 or H3K4me2 peptide (residues 1-12 of H3) into the ASH1L$_{PHD}$ NMR sampl led to large chemical shift perturbations (CSPs) in $^1H$,$^{15}N$ heteronuclear single quantum coherence (HSQC) spectra of ASH1L$_{PHD}$ (Fig. 3b). These changes were in the intermediate exchange regime on the NMR

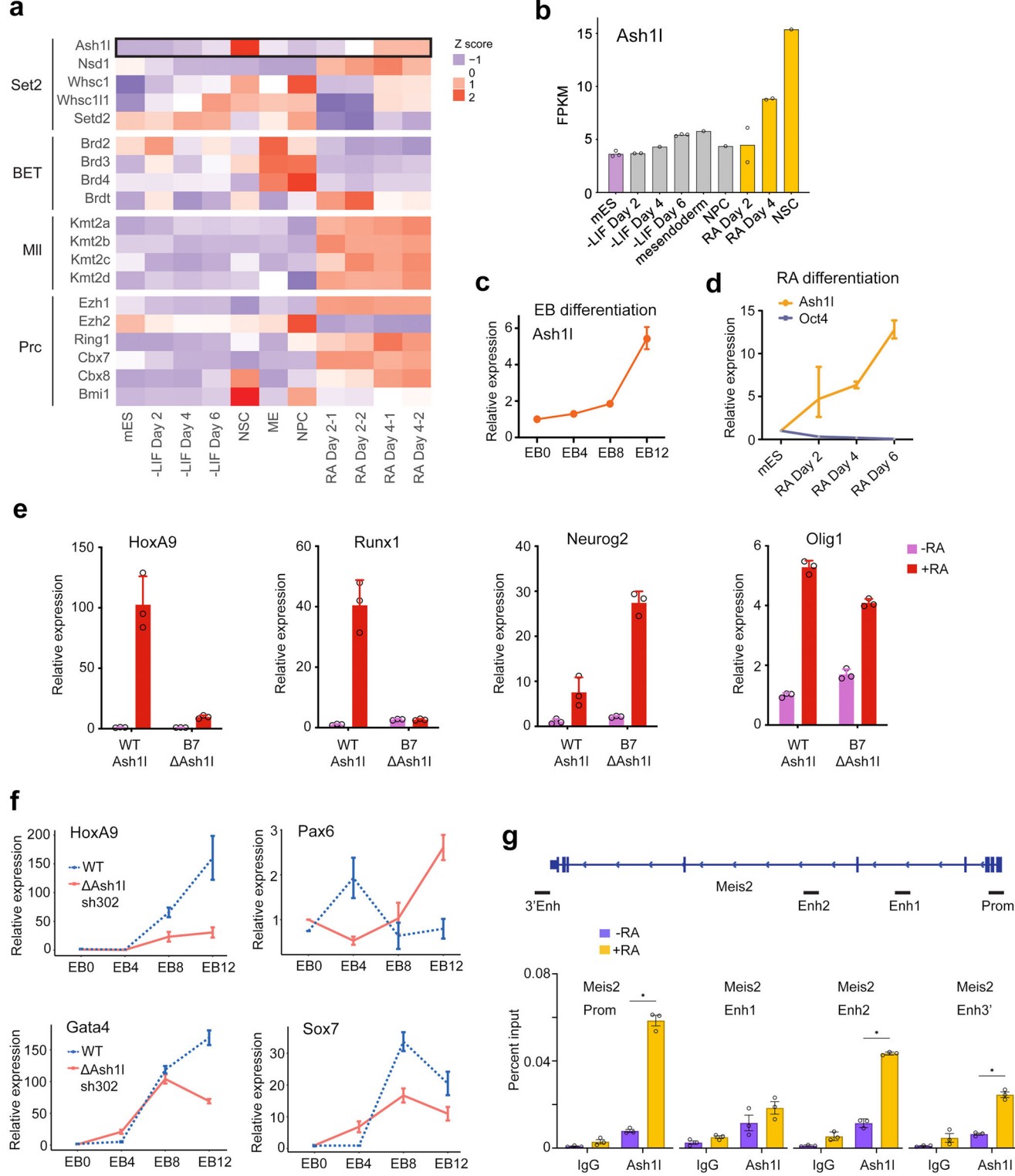

**Fig. 1 | Ash1L is essential for mouse ES cell differentiation. a** Heatmap of RNA-seq data from[19] showing relative expression levels of selected families of genes in ES cells differentiated under various conditions, as indicated. Expression values are normalized to each gene. **b** Expression levels of Ash1l in ES cells upon differentiation under various conditions as in **a**. Values are plotted in Fragments per Kilobase of Exon per Megabase (FPKM). **c**, **d** qPCR analysis of transcriptional expression of Ash1l in mouse ES cells following differentiation as embryoid bodies (EBs) (**c**) or by retinoic acid (**d**) in a time course, as indicated. Error bars represent mean from three replicates ± SEM. **e** Relative expression levels of target differentiation markers in wildtype and ΔAsh1l CRISPR edited (B7 clone) mES cells 2 days post retinoic acid induced differentiation, as measured by qPCR. Error bars represent mean from three replicates ± SEM. **f** Relative expression of target differentiation genes during embryoid body development in WT or ΔAsh1l mES cells transfected with shLuc or shAsh1l, respectively. Error bars represent mean from three replicates ± SEM. **g** Ash1l occupancy at the *Meis2* enhancer and promoter sites in mES cells pre- and post-RA-induced differentiation, as shown by ChIP-qPCR analysis. * denotes $p < 0.001$ using an unpaired t test (two-sided). Error bars represent mean from three replicates ± SEM. Source data are provided with this paper.

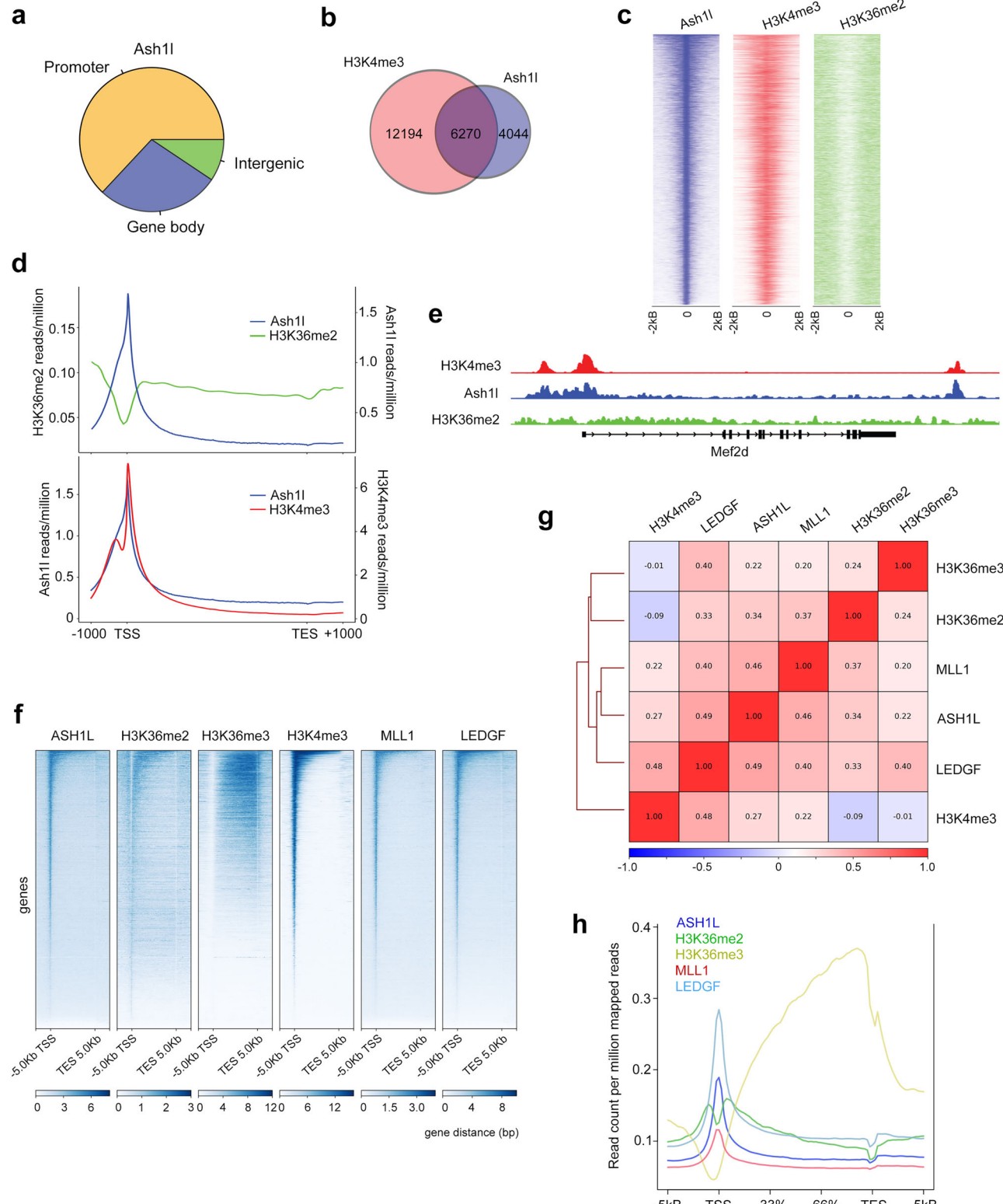

**Fig. 2 | ASH1L regulates gene expression and co-localizes with H3K4me3.**
**a** Genomic distribution of Ash1l peaks in differentiated mouse ES cells as assessed by ChIP-seq. **b** Venn diagram showing co-occupancy of Ash1l and H3K4me3 peaks in mouse ES cells, as determined by ChIP-seq. **c** Heatmaps of Ash1l, H3K4me3, and H3K36me2 occupancy across Ash1l peaks in mouse ES cells, aligned to decreasing intensity of Ash1l peaks. **d** Plots of H3K36me2 and H3K4me3 occupancy at Ash1l target genes in mouse ES cells. **e** Ash1l, H3K4me3, and H3K36me2 occupancies at

the *Mef2d* locus observed in the ChIP-seq data from mouse ES cells. **f** Peak-centered analysis showing ChIP-seq density heat maps of whole gene region occupancies (starting 5 kb upstream of TSS and ending 5 kb downstream of TES) for the indicated proteins and histone marks in human MV4-11 cells. **g** Pearson correlation of overlapping distribution profiles for the indicated proteins and histone marks in MV4-11 cells. **h** Occupancy of the indicated proteins and PTMs at ASH1L target genes in MV4-11 cells.

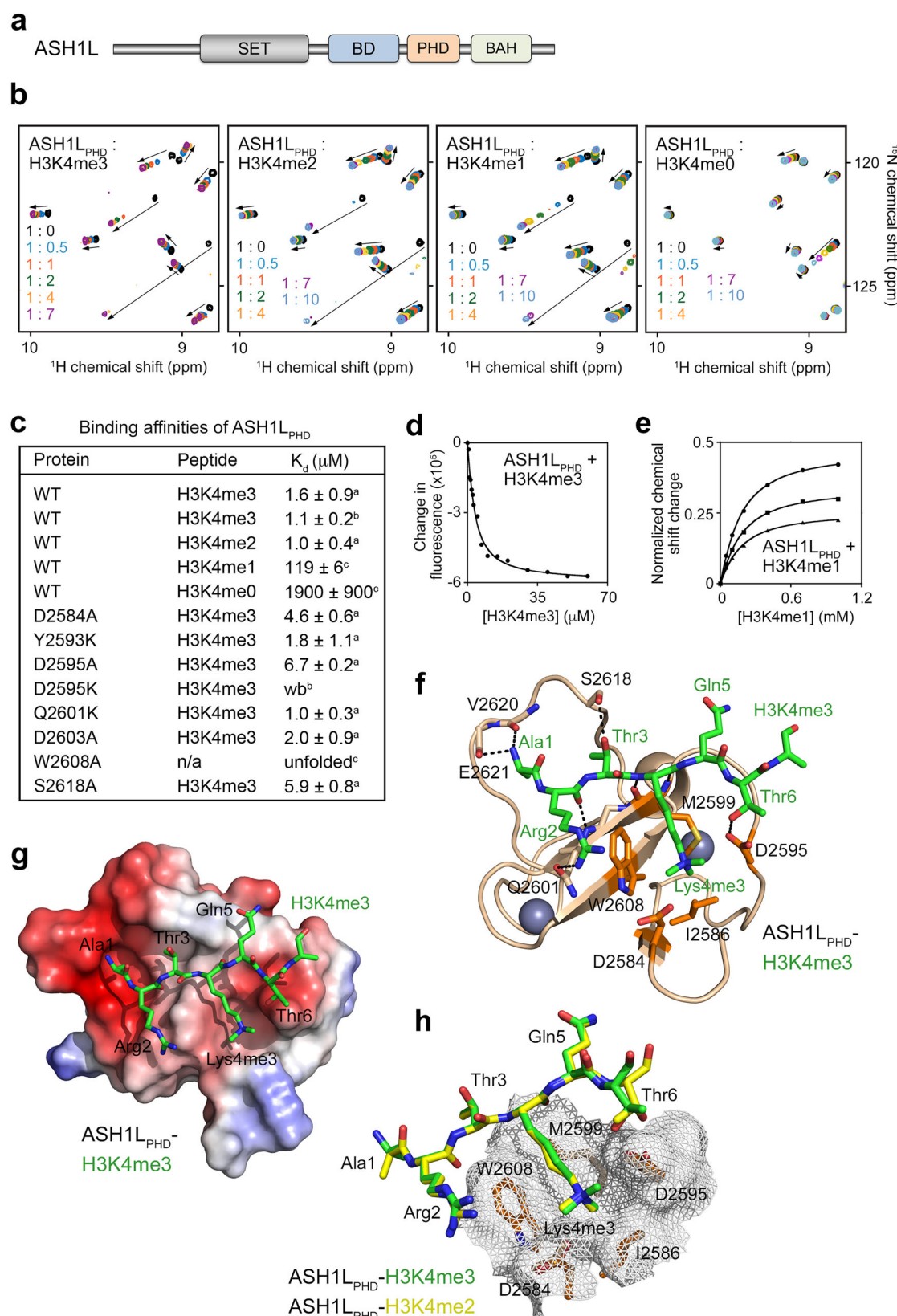

**c** Binding affinities of ASH1L$_{PHD}$

| Protein | Peptide | $K_d$ (µM) |
|---------|---------|------------|
| WT | H3K4me3 | 1.6 ± 0.9[a] |
| WT | H3K4me3 | 1.1 ± 0.2[b] |
| WT | H3K4me2 | 1.0 ± 0.4[a] |
| WT | H3K4me1 | 119 ± 6[c] |
| WT | H3K4me0 | 1900 ± 900[c] |
| D2584A | H3K4me3 | 4.6 ± 0.6[a] |
| Y2593K | H3K4me3 | 1.8 ± 1.1[a] |
| D2595A | H3K4me3 | 6.7 ± 0.2[a] |
| D2595K | H3K4me3 | wb[b] |
| Q2601K | H3K4me3 | 1.0 ± 0.3[a] |
| D2603A | H3K4me3 | 2.0 ± 0.9[a] |
| W2608A | n/a | unfolded[c] |
| S2618A | H3K4me3 | 5.9 ± 0.8[a] |

time scale, indicating tight binding. In agreement, a 1-2 µM binding affinity of ASH1L$_{PHD}$ to di- and trimethylated peptides was measured by tryptophan fluorescence and MST (Fig. 3c, d). We note that this affinity is in the range of binding affinities exhibited by the majority of histone binding modules[27–29], suggesting that the recognition of H3K4me2/3 by ASH1L$_{PHD}$ is physiologically relevant. In contrast, the association of

ASH1L$_{PHD}$ with H3K4me1 or H3K4me0 peptides was weaker ($K_d$s of 119 µM and 1900 µM, respectively, Fig. 3b, c, e).

**Molecular basis for the interaction of ASH1L$_{PHD}$ with H3K4me2/3**
To gain insight into the molecular mechanism of the recognition of H3K4me3/2, we co-crystallized ASH1L$_{PHD}$ with H3K4me3 and

**Fig. 3 | ASH1L$_{PHD}$ is a reader of H3K4me3. a** Schematic of ASH1L domain organization. **b** Overlays of ¹H,¹⁵N HSQC spectra of ASH1L$_{PHD}$ collected in the absence (black) and presence of the indicated molar ratios of histone H3K4me3, H3K4me2, H3K4me1 or H3K4me0 (all aa 1-12 of H3) peptides. **c** Binding affinities of wildtype and mutated ASH1L$_{PHD}$ for indicated histone H3 peptides, as measured by tryptophan fluorescence[a], MST[b] or NMR[c]. **d** Representative binding curve used to determine K$_d$ values by tryptophan fluorescence. K$_d$s are calculated as mean values +/- S.D. from three independent experiments. **e** Representative binding curves used to determine K$_d$ values by NMR. ¹H/¹⁵N Normalization equation is shown in NMR methods. K$_d$s are represented as mean values +/- S.D. The experiment was performed once. **f** A ribbon diagram of the crystal structure of ASH1L$_{PHD}$ (wheat) in complex with H3K4me3 peptide (green sticks). Zinc ions are shown as gray spheres and hydrogen bonds are shown as dashed lines. The trimethylammonium binding cage residues of ASH1L$_{PHD}$ are shown as sticks and colored orange. **g** Electrostatic surface potential of the ASH1L$_{PHD}$ bound to H3K4me3 peptide (green sticks). Electrostatic potential ranging from positive;blue (+100 kT/e) to negative;red (−100 kT/e) generated with PyMol vacuum electrostatics. **h** Overlay of the crystal structures of the H3K4me3-bound and H3K4me2-bound ASH1L$_{PHD}$ (also see Supplementary Fig. 5). The methyllysine binding cage surface is represented in mesh. The H3K4me3 and H3K4me2 peptides are green and yellow, respectively.

H3K4me2 peptides and determined a 1.3 Å resolution crystal structure of the ASH1L$_{PHD}$:H3K4me3 complex and a 1.9 Å resolution crystal structure of the ASH1L$_{PHD}$:H3K4me2 complex (Fig. 3f–h, Supplementary Fig. 5, and Supplementary Table 1). The ASH1L$_{PHD}$ fold is stabilized by two zinc-binding clusters and a twisted double-stranded anti-parallel β-sheet. The H3K4me3 peptide occupies an extended, shallow, negatively charged binding site of the domain. Backbone chains of histone residues R2-K4me3 pair with the existing β-sheet of ASH1L$_{PHD}$, forming the third anti-parallel β-strand, and the side chains of the peptide residues make additional intermolecular contacts. These include hydrogen bonds between the guanidino group of R2 and the side chain carbonyl of Q2601, between the hydroxyl group of T3 and the hydroxyl group of S2618, and between the hydroxyl group of T6 and the carboxyl group of D2595. Additionally, the N-terminal amino group of A1 of the peptide is restrained through the hydrogen bonding contacts with the backbone carbonyl groups of V2620 and E2621 of the protein. The trimethylated K4 is bound in the hydrophobic cage, formed by I2586, M2599 and W2608 and surrounded by the negatively charged residues D2584 and D2595 (Fig. 3h). W2608 makes a cation−π contact with the trimethylammonium group of K4 and is critical for structural stability of ASH1L$_{PHD}$, as mutation of W2608 to alanine disrupts the protein fold (Fig. 3c and Supplementary Fig. 6). All other mutants tested were folded, and while the S2618A slightly decreased binding to H3K4me3, Y2693K, Q2601K and D2603A mutations did not affect it (Fig. 3c and Supplementary Fig. 6). The D2595K mutant lost the ability to interact with H3K4me3, likely due to the electrostatic repulsion of the positively charged K4me3 group (Fig. 3c and Supplementary Fig. 6). Substitution of D2595 or D2584 with alanine decreased binding affinity to -5−7 µM, indicating the importance of the hydrogen bond of D2595 with T6 and favorable electrostatic contacts of D2595 and D2584 with K4me3 for the complex formation (Fig. 3c and Supplementary Fig. 5). The structure of the ASH1L$_{PHD}$-H3K4me3 complex overlays well with the structure of the ASH1L$_{PHD}$-H3K4me2 complex (rmsd of 0.1 Å), as well as with the structure of the ING2$_{PHD}$-H3K4me3 complex (rmsd of 1 Å)[23], pointing to a conserved mode of the methyllysine recognition (Fig. 3h and Supplementary Fig. 7).

## ASH1L$_{BD}$ has DNA binding activity

To explore the relationship between ASH1L$_{PHD}$ and the neighboring ASH1L$_{BD}$, we determined the solution structure of the tandem domain ASH1L$_{BD-PHD}$ in complex with H3K4me2 peptide by NMR (Fig. 4a–c, Supplementary Fig. 8, and Supplementary Table 2). A total of 2,645 nuclear Overhauser effect (NOE)-derived distance and torsion angle restraints were used to obtain an ensemble of the 20 lowest energy NMR structures. The structures of both ASH1L$_{BD}$ and ASH1L$_{PHD}$ were well defined (a 26-residue flexible linker between the domains was excluded from structure calculations) and showed canonical folds of bromodomains and PHD fingers and inter-domain contacts (Fig. 4b). ASH1L$_{BD}$ adopts a left-handed four-helical bundle (helices α$_Z$, α$_A$, α$_B$ and α$_C$) with its α$_B$ and α$_C$ helices facing ASH1L$_{PHD}$, which has the distinctive cross-braced zinc finger topology. Like in the ASH1L$_{PHD}$:H3K4me2 complex, in the ASH1L$_{BD-PHD}$:H3K4me2 complex, the peptide forms a third antiparallel β strand to the existing β-sheet of ASH1L$_{PHD}$ (Fig. 4c). Overall, the two structures superimpose well (rmsd of 0.9 Å), indicating that no major conformational change occurs in ASH1L$_{PHD}$ when it is linked to ASH1L$_{BD}$.

Previous studies have identified a small set of histone acetyllysine sites for ASH1L$_{BD}$, with H3K56ac being the most promising[30]. In support, the H3K56ac peptide caused resonance changes, albeit very small, in the ¹H,¹⁵N HSQC spectra of ASH1L$_{BD}$ implying that the binding is weak (Fig. 4d). However, ASH1L$_{BD}$ lacks a conserved asparagine in the acetyllysine binding site which is required for recognition of acetylated lysine residues. Indeed, mapping the most perturbed residues on the structure of ASH1L$_{BD}$ revealed a separate H3K56ac binding site close to but still outside the canonical acetyllysine binding site (Supplementary Fig. 9).

Analysis of the electrostatic surface potential of ASH1L$_{BD}$ revealed two highly positively charged surface regions located far from the H3K56ac binding site (Supplementary Fig. 9). These regions encompass residues K2477/K2478/K2479 and K2524/R2528/K2529 of ASH1L$_{BD}$ that could potentially interact with negatively charged DNA (Fig. 4b). To determine whether ASH1L$_{BD}$ is capable of binding to DNA, we mutated both regions and examined the association of WT ASH1L$_{BD}$ and K2477Q/K2478Q/K2479Q and K2524Q/R2528Q/K2529Q mutants of ASH1L$_{BD}$ with 147 bp 601 DNA in an electrophoretic mobility shift assay (EMSA) (Fig. 4e–g). Increasing amounts of ASH1L$_{BD}$ were incubated with 601 DNA, and the reaction mixtures were resolved on a native polyacrylamide gel. A gradual increase in ASH1L$_{BD}$ concentration resulted in the shift of the DNA band, indicating that wild type ASH1L$_{BD}$ forms a complex with DNA (Fig. 4e). The K2524Q/R2528Q/K2529Q mutant of ASH1L$_{BD}$ however lost its ability to bind DNA, and binding of the K2477Q/K2478Q/K2479Q mutant was diminished (Fig. 4f, g). These data suggest that both positively charged regions of ASH1L$_{BD}$ are involved in the interaction with DNA.

## ASH1L$_{BAH}$ and ASH1L$_{PHD}$ form an integrated module

To characterize the relationship between ASH1L$_{PHD}$ and the following ASH1L$_{BAH}$, we superimposed ¹H,¹⁵N transverse relaxation optimized spectroscopy (TROSY) spectra of isolated ASH1L$_{PHD}$ and the double domain ASH1L$_{PHD-BAH}$ (Fig. 5a). Notably, cross-peaks in the two spectra were not superimposable, suggesting that either the two domains of ASH1L interact or conformational changes occur when they are connected. To elucidate the interplay, we determined a 2.4 Å crystal structure of ASH1L$_{PHD-BAH}$ (Fig. 5b–e). Although ASH1L$_{PHD-BAH}$ was crystallized in the presence of H3K4me3 peptide, we did not observe electron density for the peptide. The structure of ASH1L$_{PHD-BAH}$ shows the formation of an integrated module with the extensive interface between ASH1L$_{PHD}$ and ASH1L$_{BAH}$. The considerable size of this interface suggests that ASH1L$_{PHD}$ supports the BAH fold and may explain as to why we were unable to produce an isolated folded ASH1L$_{BAH}$.

A large network of inter-domain hydrogen bonding, hydrophobic, and electrostatic contacts stabilize the ASH1L$_{PHD-BAH}$ structure (Fig. 5c). The guanidino group of R2642 is hydrogen bonded to the backbone carbonyl group of C2650, whereas the

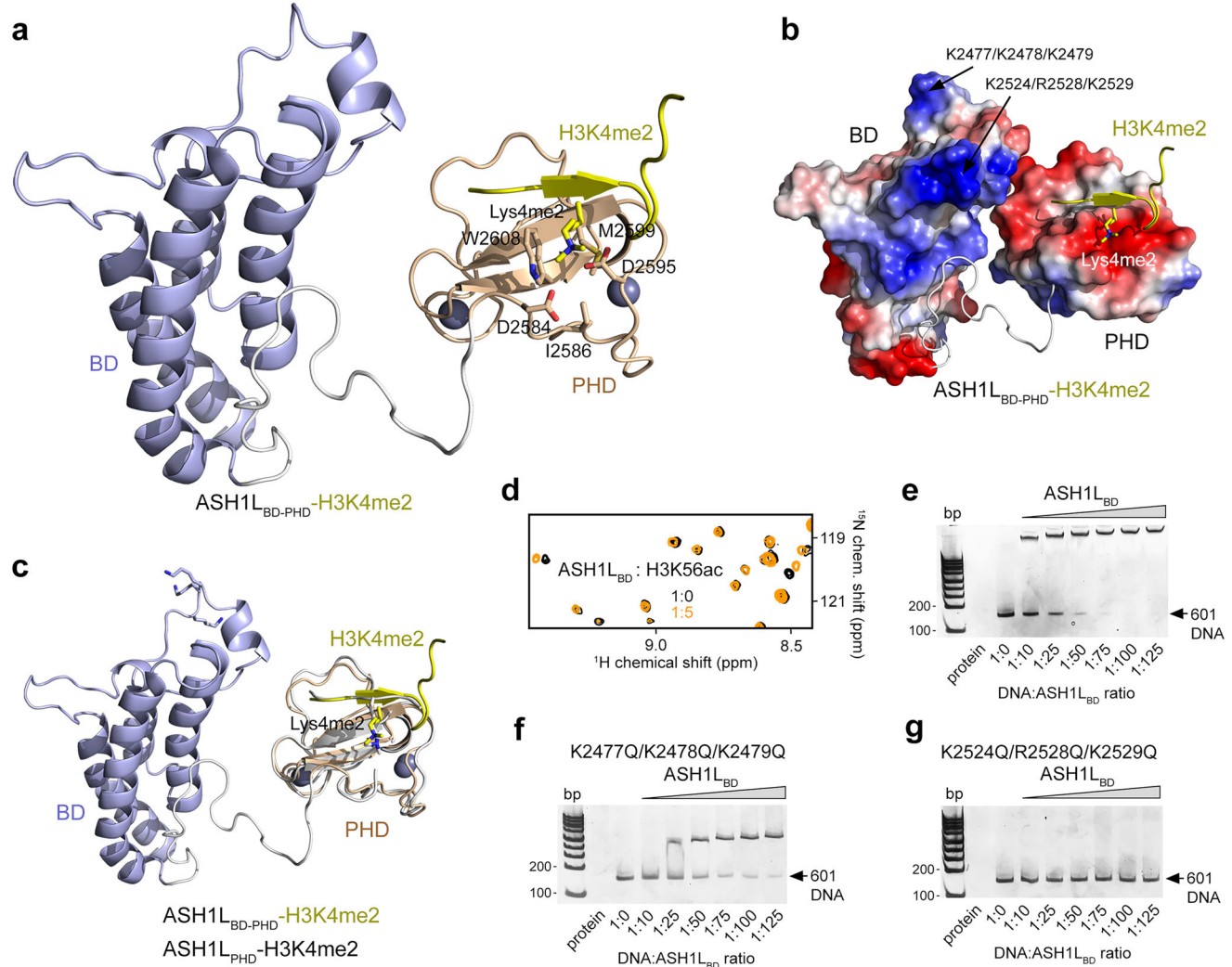

**Fig. 4 | DNA and histone binding activities of ASH1L_BD. a** Ribbon diagram of the NMR solution structure of ASH1L_BD-PHD (BD, blue and PHD, wheat) in complex with H3K4me2 peptide (yellow). **b** Electrostatic surface potential of ASH1L_BD-PHD bound to H3K4me2 peptide (yellow). Unstructured linker region between the domains is shown as gray loop. Electrostatic potential ranging from positive;blue (+100 kT/e) to negative;red (−100 kT/e) generated with PyMol vacuum electrostatics. **c** Overlay of the NMR solution structure of ASH1L_BD-PHD (BD, blue and PHD, wheat) bound to H3K4me2 peptide (yellow) with the crystal structure of ASH1L_PHD (gray) bound to H3K4me2 peptide (gray). **d** Overlay of $^1$H,$^{15}$N HSQC spectra of ASH1L_BD in the absence (black) or presence (orange) of H3K56ac peptide. **e–g** EMSA with 147-bp 601 DNA in the presence of increasing amounts of ASH1L_BD, WT and the indicated mutants. Source data are provided with this paper.

backbone amino group of M2639 donates a hydrogen bond to the backbone carbonyl of Y2652. The backbone amino and carbonyl groups of V2637 are engaged in hydrogen bonds with the backbone carbonyl group of C2655 and the backbone amino group of I2654, respectively. The side chain carboxyl group of E2636 is restrained through a salt bridge and hydrogen bonds with the side chain amino group of K2709, the side chain guanidino group of R2715, and the backbone amino group of L2657. The guanidino groups of R2635 and R2664 form hydrogen bonds with the backbone carbonyl groups of D2660 and C2590, respectively. The side chain amino group of K2807 is hydrogen bonded to the side chain carboxyl group of D2595, whereas the side chain hydroxyl moiety of Y2809 makes a hydrogen bond with the backbone amino group of I2586.Overlay of the crystal structures of ASH1L_PHD-BAH and H3K4me3-bound ASH1L_PHD showed that the ASH1L_PHD fold superimposes very well and the H3K4me3 binding site of ASH1L_PHD is not occluded by ASH1L_BAH (Fig. 5d). In agreement, binding affinity of ASH1L_PHD-BAH for H3K4me3, was similar to the binding affinity of ASH1L_PHD for the same H3K4me3 peptide (Figs. 3c and 5f).

## ASH1L_BAH is a DNA binding module with preference for linker DNA

The electrostatic surface potential of the ASH1L_PHD-BAH structure showed a large positively charged surface of ASH1L_BAH positioned next to the negatively charged H3K4me3-binding site of ASH1_PHD and thus suggested that ASH1L_BAH might bind DNA (Fig. 5e). EMSA assays confirmed that ASH1L_PHD-BAH forms a tight complex with 601 DNA, and the DNA binding activity was further increased when ASH1L_PHD-BAH was linked to ASH1L_BD in ASH1L_BD-PHD-BAH (Fig. 5g and Supplementary Fig. 10a). ASH1L_PHD-BAH also formed a tight complex with the H3K4me3-containing 147 bp nucleosome core particle (H3K4me3-NCP) (Fig. 5h). Binding to unmodified 147 bp NCP was decreased, pointing to the contribution of the interaction of ASH1L_PHD in ASH1L_PHD-BAH with H3K4me3, whereas binding to unmodified 187 bp NCP was increased compared to its binding to unmodified 147 bp NCP, indicating that ASH1L_BAH in ASH1L_PHD-BAH prefers extra-nucleosomal linker DNA, and this preference was retained in ASH1L_BD-PHD-BAH (Supplementary Figs. 10b and 11). These data demonstrate the importance of both interactions of ASH1L_PHD with H3K4me3 and ASH1L_BAH with the linker DNA for binding of ASH1L to the nucleosome.

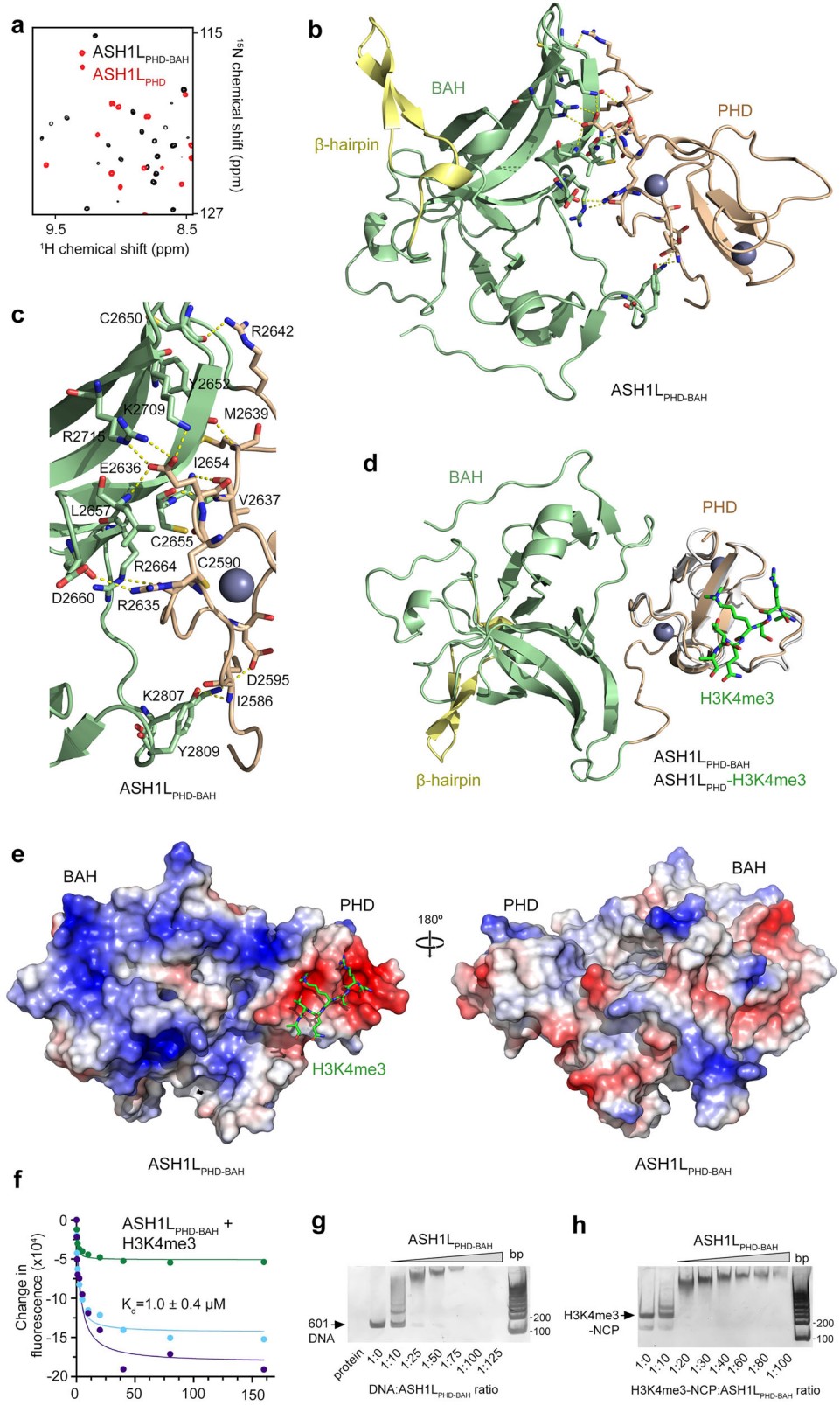

## ASH1L$_{PHD}$-H3K4me3 interaction negatively regulates H3K36 dimethylation

*Drosophila* Ash1 has been shown to exist in a complex with the Mrg15 and Nurf55 subunits, with the former stimulating the catalytic activity of Ash1[31,32]. In support, MRG15 increases histone methyltransferase (HMT) activity of human ASH1L$_{SET-BD-PHD}$ on native chromatin substrate (Supplementary Fig. 12a). Interestingly, titration of an isolated unlabeled ASH1L$_{SET}$ domain into $^{15}$N-labeled ASH1L$_{BD}$ or $^{15}$N-labeled ASH1L$_{PHD}$ caused CSPs in ASH1L$_{BD}$ but not in ASH1L$_{PHD}$ (Fig. 6a, b). These data indicate that ASH1L$_{SET}$ is in direct contact with ASH1L$_{BD}$, however it does not interact with ASH1L$_{PHD}$.

**Fig. 5 | ASH1L$_{BAH}$ fuses with ASH1L$_{PHD}$ and binds DNA. a** Overlay of ${}^1$H,${}^{15}$N TROSY spectra of ASH1L$_{PHD-BAH}$ (black) and ASH1L$_{PHD}$ (red). **b** A ribbon diagram of the crystal structure of ASH1L$_{PHD-BAH}$ (PHD, wheat; BAH, green; and a β-hairpin insertion, yellow). Zinc ions are shown as gray spheres, and dashed lines indicate hydrogen bonds. **c** A zoom-in view of the PHD-BAH interface. Dashed lines indicate hydrogen bonds, and the interacting residues (sticks) are labeled. **d** Overlay of crystal structures of ASH1L$_{PHD-BAH}$ (PHD, wheat; BAH, green; and a β-hairpin insertion, yellow) and ASH1L$_{PHD}$ (gray) bound to H3K4me3 peptide (green).

**e** Electrostatic surface potential of ASH1L$_{PHD-BAH}$. Histone H3K4me3 (green sticks) positioned based on overlay in **d**. Electrostatic potential ranging from positive;blue (+100 kT/e) to negative;red (−100 kT/e) generated with PyMol vacuum electrostatics. **f** Representative binding curves used to determine K$_d$ values of ASH1L$_{PHD-BAH}$ for H3K4me3 peptide by tryptophan fluorescence. K$_d$ was calculated as a mean value +/- S.D. from three independent experiments. **g, h** EMSA with 147 bp 601 DNA (**g**) or 147 bp H3K4me3-NCP (**h**) in the presence of increasing amounts of ASH1L$_{PHD-BAH}$. Source data are provided with this paper.

To determine the effect of the ASH1L$_{PHD}$-H3K4me3 interaction on the catalytic function of ASH1L, we tested ASH1L$_{SET-BD-PHD-BAH}$ on recombinant H3K4me3-NCP and unmodified NCP (Fig. 6c). Unexpectedly, we found that the catalytic activity of ASH1L$_{SET-BD-PHD-BAH}$ was reduced in the presence of H3K4me3, i.e., on H3K4me3-NCP as compared to unmodified NCP. Similarly, the level of H3K36 mono- and dimethylation produced by ASH1L$_{SET-BD-PHD}$ on H3K4me3-NCP was lower than on unmodified NCP (Fig. 6d, Supplementary Fig. 12b). Furthermore, impairing binding of ASH1L$_{PHD}$ to H3K4me3 by mutating D2595 led to an increase in the methyltransferase activity of the ASH1L$_{SET-BD-PHD}$ D2595K mutant compared to the catalytic activity of WT ASH1L$_{SET-BD-PHD}$ on H3K4me3-NCP (Fig. 6e). Disruption of the ASH1L$_{PHD}$-H3K4me3 interaction either in the presence or absence of MRG15 resulted in a more robust catalytic activity. Together, these data indicate that the recognition of H3K4me3 by ASH1L$_{PHD}$ is inhibitory to H3K36 methylation by ASH1L$_{SET}$. This inhibition could potentially explain the negative correlation observed between occupancy of ASH1L at the H3K4me3-enriched TSS sites and H3K36me2 in two tested cell lines.

## ASH1L is implicated in cancer

According to TCGA, the *ASH1L* gene is frequently altered in human cancers. Mutations and amplifications are particularly prevalent in lung cancer, with *ASH1L* alterations being found in ~15% of lung adenocarcinoma (cBioPortal). We examined the ASH1L protein expression level in five lung adenocarcinoma cell lines by western blot. As shown in Supplementary Fig. 13a, ASH1L was highly overexpressed in A549 cells. RNA-seq experiments using A549 cells in which endogenous ASH1L was depleted by two shRNAs showed that 541 genes were downregulated and 398 genes were upregulated in ASH1L knockdown cells (FC > 2, FDR < 0.05, Fig. 6f and Supplementary Fig. 13b). Ingenuity Pathway Analysis and GO analyses revealed that downregulated genes are implicated in the cell cycle, DNA replication and cell death and survival – the vital cellular pathways, misregulation of which can lead to the development of cancer and other human diseases (Fig. 6g and Supplementary Data 1). In agreement, ASH1L depletion markedly reduced the growth of A549 cancer cells (Fig. 6h), and lung adenocarcinoma patients with low levels of ASH1L have better survival (Fig. 6i). Over a dozen mutations associated with cancer have been identified in ASH1L$_{PHD}$ (Cosmic). We generated cancer-relevant R2587C, L2624F and D2629H mutants of ASH1L$_{PHD}$ and found that binding of these mutants, especially of R2587C, to H3K4me3 peptide was diminished, suggesting that the impaired function of ASH1L$_{PHD}$ could be potentially linked to aberrant activity of ASH1L (Supplementary Fig. 14). Our findings support the role of ASH1L in oncogenesis and may provide further insights into a rational design of strategies to target ASH1L.

## Discussion

Dimethylation of H3K36 is critical for transcriptional regulation and DNA damage repair. In mammals, H3K36me2 is catalyzed by a set of methyltransferases, including ASH1L, a large protein that remains poorly characterized, likely due to its size. Although progress has been made toward understanding of the catalytic activity and autoregulation of ASH1L$_{SET}$, much remains to be learned about functions of the followed ASH1L$_{BD}$, ASH1L$_{PHD}$ and ASH1L$_{BAH}$ domains. In this work, we have identified the biological roles of ASH1L$_{BD}$, ASH1L$_{PHD}$ and ASH1L$_{BAH}$, structurally characterized these domains, determined their

mechanisms of action, and explored functional crosstalk between these modules (Fig. 6j, k). We found that ASH1L$_{PHD}$ recognizes H3K4me2/3, whereas the neighboring ASH1L$_{BD}$ and ASH1L$_{BAH}$ have DNA binding activities. We further demonstrate that ASH1L$_{BD}$ directly interacts with ASH1L$_{SET}$ and can bind H3K56ac, albeit weakly. The DNA binding function of ASH1L$_{BAH}$ is a driving force for the association of ASH1L with the linker DNA in the nucleosome, and the large interface with ASH1L$_{PHD}$ stabilizes the ASH1L$_{BAH}$ fold, merging two domains into a single module. Overlay of the structures of H3K4me2-bound ASH1L$_{BD-PHD}$ and ASH1L$_{PHD-BAH}$ shows that ASH1L$_{PHD}$ is surrounded by ASH1L$_{BD}$ from one side and by ASH1L$_{BAH}$ from another (Fig. 6k). Notably, electrostatic surface potential of the domains, shown in Supplementary Fig. 11e, suggests a fine-tuned balance of electrostatic contacts with the nucleosome as well as between the ASH1L domains that can be in turn regulated through binding of ligands or the linkers connecting these domains, therefore, it will be essential in future studies to define how the entire C-terminal region of ASH1L, including ASH1L$_{SET}$, engages the nucleosome and nucleosomal arrays.

ASH1L acts synergistically with MLL1, and both are required for the effective expression of the HOX genes[4,12,15,20,33], however, the relationship between H3K36me2 generated by ASH1L and H3K4me3 generated by MLL1 remains unclear. Our genomic data analysis of two cell lines reveals that ASH1L and H3K4me3 co-occupy TSS where H3K36me2 is depleted, suggesting that this methyltransferase may not be fully active at the H3K4me3-rich sites. The finding that the interaction of ASH1L$_{PHD}$ with H3K4me3 is inhibitory to the catalytic activity of ASH1L could offer a possible explanation for such a decrease. DNA binding and/or direct association of ASH1L$_{BD}$ with ASH1L$_{SET}$ could play a role in autoinhibition of ASH1L$_{SET}$ by its autoinhibitory loop, which in turn could be affected or regulated via the binding of ASH1L$_{PHD}$ to H3K4me3. Furthermore, binding of the integrated ASH1L$_{PHD-BAH}$ module to H3K4me3 and the linker DNA could impede priming of the catalytic ASH1L$_{SET}$ at H3K36. It is also possible that the entire ligand-bound ASH1L$_{BD-PHD-BAH}$ allosterically impacts the orientation of the autoinhibitory loop of ASH1L$_{SET}$. Overall, our current model suggests that ASH1L senses local epigenetic environment through its ASH1L$_{PHD}$ and produces H3K36me2 at the sites with low levels of H3K4me3. Of note, many oncogenic activities of ASH1L are attributed to its C-terminal region, which is sufficient to induce the HOX genes expression and forms the translocated onco-fusion with NUP98[12,20,34,35]. Inhibition of ASH1L has been proposed as a potential therapeutic approach, and small molecule inhibitors of ASH1L$_{SET}$ have already shown promising results in blocking proliferation of cancer cells[15]. Our findings provide molecular and structural insights that may help in a rational design of novel diverse strategies to therapeutically target ASH1L.

## Methods

### Mouse ES cell culture

J1 ESCs were cultured feeder free on 0.1% gelatin-coated plates in ESC culture medium (Dulbecco's modified Eagle's medium (Hi-glucose), supplemented with 15% fetal bovine serum (Corning), non-essential amino acids, L-glutamine, β-mercaptoethanol, 1% penicillin/streptomycin, sodium pyruvate and 1000 U/mL leukemia inhibitory factor (LIF) (Ebioscience, 34-8521-82). For embryoid body formation, mouse ES cells were dissociated from gelatin-coated plates and seeded on non-adherent cell culture dishes (Greiner) at a density of $1 \times 10^5$ cells/mL in 10 mL of differentiation medium (ESC culture medium without LIF).

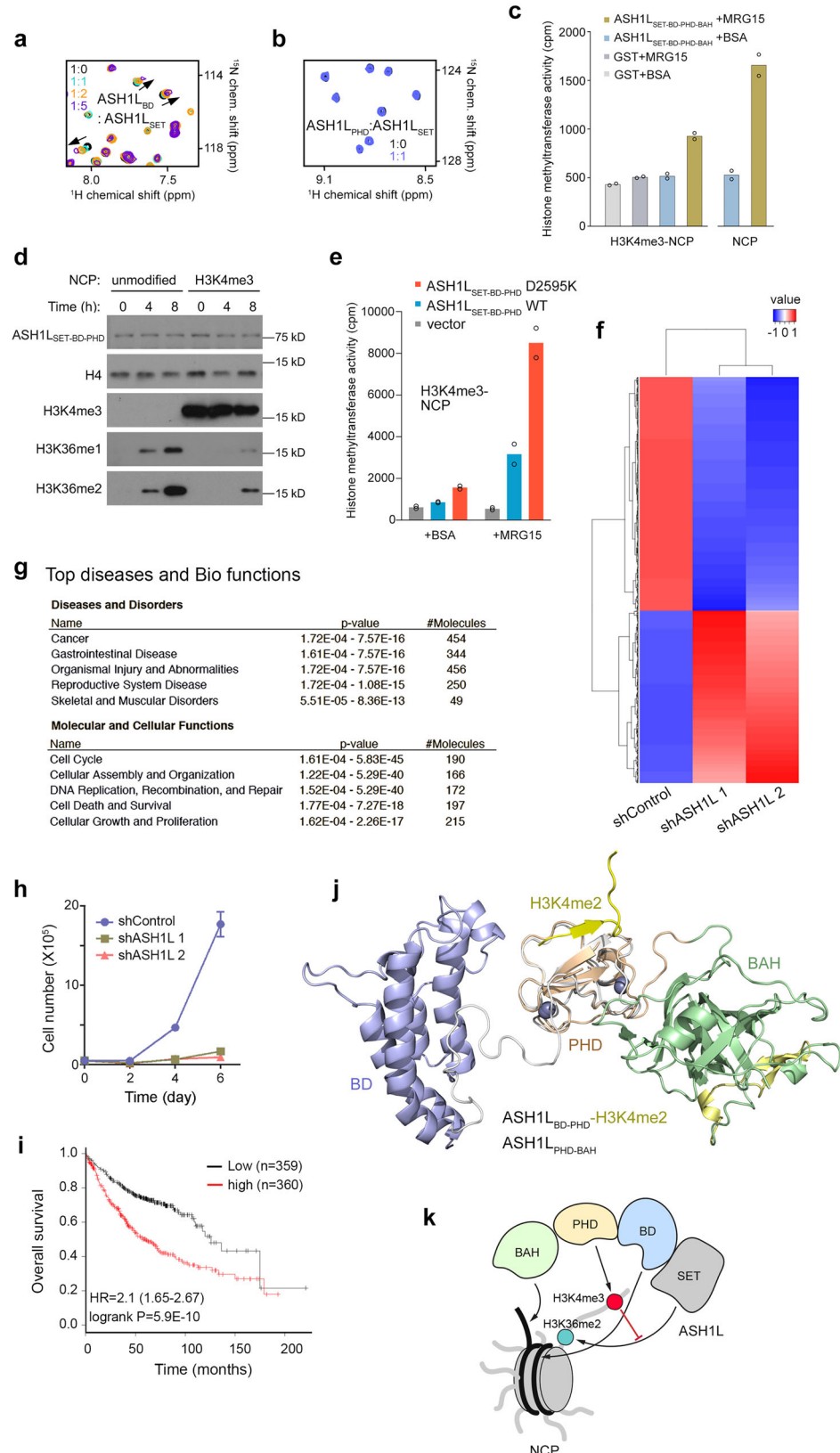

g    Top diseases and Bio functions

**Diseases and Disorders**

| Name | p-value | #Molecules |
|---|---|---|
| Cancer | 1.72E-04 - 7.57E-16 | 454 |
| Gastrointestinal Disease | 1.61E-04 - 7.57E-16 | 344 |
| Organismal Injury and Abnormalities | 1.72E-04 - 7.57E-16 | 456 |
| Reproductive System Disease | 1.72E-04 - 1.08E-15 | 250 |
| Skeletal and Muscular Disorders | 5.51E-05 - 8.36E-13 | 49 |

**Molecular and Cellular Functions**

| Name | p-value | #Molecules |
|---|---|---|
| Cell Cycle | 1.61E-04 - 5.83E-45 | 190 |
| Cellular Assembly and Organization | 1.22E-04 - 5.29E-40 | 166 |
| DNA Replication, Recombination, and Repair | 1.52E-04 - 5.29E-40 | 172 |
| Cell Death and Survival | 1.77E-04 - 7.27E-18 | 197 |
| Cellular Growth and Proliferation | 1.62E-04 - 2.26E-17 | 215 |

Media was exchanged every 48 h and cells were collected at indicated time course. For neuronal differentiation, cells were plated on gelatin-coated plates at a density of $1 \times 10^4$ cells/cm$^2$ in ESC culture medium. The following day, medium was exchanged to differentiation medium +1 µM retinoic acid (RA) (Sigma) to induce differentiation. Medium was exchanged every 48 h for indicated time course.

**Real time quantitative PCR (RT-qPCR) and ChIP-qPCR**
Total RNA was extracted with RNeasy Mini Kit (QIAGEN) and reverse transcribed using Superscript III Reverse Transcriptase (Life Technologies). qPCR analysis was performed using Brilliant III Ultra Fast SYBR Green QPCR Master Mix (Agilent Technologies). Gene expression data were normalized against Hprt/Gapdh and represented as fold change

**Fig. 6 | Binding of ASH1L$_{PHD}$ to H3K4me3 inhibits enzymatic activity of ASH1L.** **a** Overlay of $^{1}$H,$^{15}$N HSQC spectra of ASH1L$_{BD}$ in the absence (black) or presence of the indicated molar ratios of ASH1L$_{SET}$. **b** Overlay of $^{1}$H,$^{15}$N HSQC spectra of ASH1L$_{PHD}$ in the absence (black) or presence (blue) of ASH1L$_{SET}$. **c** Liquid histone methyltransferase (HMT) assays of GST-ASH1L$_{SET-BD-PHD-BAH}$ on recombinant nucleosomes carrying H3K4me3 or unmodified NCP. GST-ASH1L$_{SET-BD-PHD-BAH}$ was pre-incubated with substrate and MRG15 (or BSA) prior addition of $^{3}$H S-adenosyl methionine. GST, control. Experiment shown was performed in duplicate and was repeated at least three times with different preps of recombinant GST-ASH1L$_{SET-BD-PHD-BAH}$ with similar results. **d** Western blot analysis of in vitro KMT assays of recombinant GST-ASH1L$_{SET-BD-PHD}$ on unmodified and H3K4me3-nucleosomes. **e** Liquid HMT assays of recombinant GST-ASH1L$_{SET-BD-PHD}$ on H3K4me3-nucleosomes in the absence (BSA) or presence of MRG15. Experiment shown was performed in duplicate and was repeated at least three times with different preps of recombinant proteins with similar results. **f** Heatmap of gene expression profiles in human lung adenocarcinoma A549 cells transfected with control (shNT) or shASH1L. Blue and red colors indicate down and up regulation of genes, respectively. **g** IPA analysis of the downregulated 541 genes in ASH1L knockdown cells. The top five hits of each category are listed. Statistical test used was one-sided. **h** Cell proliferation assays of the control (shControl) and ASH1L knockdown A549 cells. Live cells were counted over a 6-day time course. Error bars represent mean from three replicates ± SEM. **i** Kaplan-Meier analysis of human lung adenocarcinoma patients with low or high *ASH1L* gene expression levels. **j** Overlay of the structures of the H3K4me2-bound ASH1L$_{BD-PHD}$ and ASH1L$_{PHD-BAH}$. **k** A model of the functional cooperation between ASH1L domains in the context of the nucleosome. DNA and histone tails are shown as black and gray curves, respectively. Histone H3 modifications are indicated by colored circles and labeled. Source data are provided with this paper.

relative to the control. The relative expression values are shown as mean ± S.D. For ChIP-qPCR, chromatin occupancy was calculated as the ratio of the amount of immunoprecipitated DNA to that of the input sample (% input).

## CRISPR mediated knockout

CRISPR/Cas9 mediated editing was performed essentially as described[36]. The designed gRNA (CACCGATCCTGCCTATTTCGAAGC) was cloned into the pSpCas9(BB)-2A-GFP (PX458) plasmid (a gift from Feng Zhang) and miniprepped using standard DNA miniprep procedures (Qiagen). J1 mES cells were transfected with sgRNAs or empty vector with Lipofectamine 2000 (ThermoFisher). Transfected cells were isolated by FACS at the Mount Sinai Flow Cytometry Core facility, and a single cell was plated per well in 96-well plates. Cells were cultured for 2-3 weeks until single colonies were visible. Cells were harvested and DNA was extracted using the QuickExtract solution (Epibio). PCR was performed using the following primers: Ash1l, F:TAGAAAGTCTGCACGGGCG, R:GC ACGAAGTTCCTCTCTCTGTT, Nestin, F:GCAGGAGAAGCAGGGTCTAC, R:CTTGGGGTCAGGAAAGCCAA, HoxA9, F:CCGGACGGCAGTTGATAGA G, R: CCAGCGTCTGGTGTTTTGTG, Pax6, F:CCGAGAAGCGGCTTTGAGA A, R:ATACGGGGCTCTGAGAACTG, Gata4, F: TCCATGTCCCAGACATTC AGT, R: TACGCGGTGATTATGTCCCC, Sox7, F:TCAGGGGACAAGAGTTC GGA, R:CCTTCCATGACTTTCCCAGCA, Pou5f1, F:AGTTGGCGTGGAGA CTTTGC, R:CAGGGCTTTCATGTCCTGG, Brachyury, F:GAGAGCGCAGG GAAGAGC, R:ACATCCTCCTGCCGTTCTTG, Nkx2-5, F:TGCAGAAGGCAG TGGAGCTGGACAAGCC, R:TGCACTTGTAGCGACGGTTCTGGAACCAG.

## Lentiviral Knockdown

shRNAs, TCCTAAGGTTAAGTCGCCCTC for shLuc and ATTGAGCAA-TAAATGACCAGC for sh302 ASH1L, were cloned into a pLKO.1-puro vector (a gift from David Root) according to standard procedures. Lentivirus was generated by transfecting psPax2 (Addgene, #12259) and VSV-G (Addgene, #8454) into HEK293T cells (Takara Bio, #632180) with Lipofectamine 2000 (ThermoFisher), in media without antibiotics. After 2–3 days, the media was harvested and spun down to pellet any debris. The supernatant was filtered and then concentrated using Amicon 30 kDa CO concentrator (Millipore). After lentivirus production, J1 mES cells were transfected with the virus in ESC culture medium supplemented with 8 µg/mL polybrene (Sigma) and cultured overnight. Stable cells were selected with 1 µg/mL puromycin (Sigma) for 24–48 h before collection for further experiments.

## ChIP-seq analysis in mouse ES cells and human MV4-11 cells

Whole-genome ChIP in J1 mouse ES cells were performed under conditions described previously[37]. J1 mouse ES cells were cultured to a density of $2 \times 10^{7}$ cells in 15 cm dishes (Greiner). 24 h after plating, media was exchanged to differentiation medium +1 µM RA (Sigma). Cells were harvested for immunoprecipitation after 48 h. Input controls were generated by sonication and purification of cross-linked chromatin without immunoprecipitation. ChIP assays were performed on the cell lysates using the following antibodies: Ash1l (Bethyl, A301-749A; 5 µg/20 million cells), H3K36me2 (Abcam, ab9049; 3 µg/3 million cells), H3K4me3 (Abcam, ab8580; 3 µg/3 million cells). ChIP-seq libraries were prepared according to the Illumina protocol and sequenced using an Illumina HiSeq. For ChIP-seq of differentiated mouse ESCs, reads were filtered using Trimmomatic v0.40 and aligned to the mouse reference genome (mm9) using Bowtie (v1.0.1) with default parameters. Wig files were generated using IGVtools count (-e 200 -w 25) and visualized in IGV. Histone peaks were called using MACS2 callpeak (-t [protein of interest] -c [input control] -g mm -q .05 --nomodel --extsize=150). For Ash1l ChIP-seq, enriched regions with fold changed >= 6 and FDR < 0.001 were called using MACS3. These identified peaks were used to create metagene heatmaps using the deepTools computeMatrix function. Bigwig files for heatmaps were generated using UCSC's wigToBigWig tool. Intersecting peaks were identified using the BEDTools intersect function and genomic distributions were determined using the HOMER suite.

Analysis of previously reported ChIP-seq datasets from the human MV4-11 cell line[4] was performed as follows. Sequencing reads were mapped to the human reference genome (hg19) using Bowtie (parameters: -v 2 -k 2 -m 1 --best --strata). Clonal reads were removed before enriched peak/region calling with MACS version 2.1.0. Enriched regions (fold change ≥ 5 over input controls) were called based on Poisson distribution model with false discovery rate ≤ 0.01 to be considered as statistically significant ChIP peaks. Downstream analyses including genome-wide localization density and average profile plots were performed by using deepTools[38].

## RNA-seq analysis of mouse EC cells

RNA-seq data reported by Yin et al.[19] was analyzed to compare RNA from undifferentiated mouse ES cells to RNA from mouse ES cells differentiated under various conditions, as described in Fig. 1. Reads were filtered using Trimmomatic v0.40 and mapped to the mouse reference genome (mm9) by TopHat (version 2.0.10). Fragments were enumerated to the UCSC reference transcriptome using the hseq-count function from the HTSeq package (version 0.6.0). Genes with <5 fragments were filtered out and gene expression changes (FC > 1.5, FDR < 0.01) were analyzed using DESeq2 (v1.42.1), with undifferentiated cells as controls. Heatmaps were generated using either the ggplot2 or pheatmap packages with z-score normalization.

## Cell proliferation, shRNA KD and RNA-seq analysis in A549 cells

Human lung cancer cell lines were maintained in RPMI or DMEM (Cellgro) supplemented with 10% fetal bovine serum (Sigma). Lentiviral-mediated shRNA transduction was performed as described previously[39]. Briefly, A549 cells were co-transfected with pMD2.G,

pPAX2 (Addgene) and pLKO-shRNA constructs. For infections, the cells were incubated with viral supernatants in the presence of 8 µg/ml polybrene. After 48 h, infected cells were selected with puromycin (2 µg/ml). Cell proliferation and colony formation assays were performed as previously described[39]. Total RNA was prepared using the RNeasy Plus kit (Qiagen) and reverse-transcribed using the First Strand Synthesis kit (Invitrogen). RNA-seq samples were sequenced using the Illumina Hiseq 2500, and raw reads were mapped to the human reference genome (hg19) by TopHat (version 2.0.10). The number of fragments in each known gene from RefSeq database (downloaded from UCSC Genome Browser on March 9, 2012) was enumerated using htseq-count from HTSeq package (version 0.6.0). Genes with less than 10 fragments in all the samples were removed, and then the normalized fragment count for each gene was calculated by R/Bioconductor package DESeq (version 1.18.0). Gene Ontology analysis was performed using Qiagen Ingenuity Pathway Analysis and the DAVID Bioinformatics Resource. Survival analyses were performed as described previously[39]. The database from The Kaplan–Meier plotter central server that consists of the expression profiling datasets of human lung adenocarcinoma patients was used in the "consolidated" survival analysis. Data were loaded into the R statistical environment for analysis. The package "overall survival" is used to calculate and plot Kaplan–Meier survival curves. P-values were calculated using logrank test. ASH1L shRNAs (shASH1L 1: TRCN 0000358527, shASH1L 2: TRCN 0000246167) were purchased from Sigma. The anti-ASH1L antibody (A301-749A, 1:1000) was purchased from Bethyl.

## Protein expression and purification
Human ASH1L constructs ASH1L$_{PHD}$ (aa 2579-2629), ASH1L$_{BD}$ (aa 2430-2564) and ASH1L$_{PHD-BAH}$ (aa 2579- 2794 and 2579-2834), ASH1L$_{BD-PHD}$ (aa 2430-2629) were cloned into pDEST15 vector with a TEV or Precission cleavage site. ASH1L$_{BD-PHD-BAH}$ (aa 2436-2833), ASH1L$_{SET-BD-PHD}$ (aa 2074-2629) and ASH1L$_{SET-BD-PHD-BAH}$ (aa 2069-2834) were in pGEX6P-1 and pMOCR vectors. Mutants of Ash1L$_{PHD}$ and Ash1L$_{BD}$ were generated following Quikchange mutagenesis protocols. Each construct was expressed in BL21 or Rosetta2 (DE3) pLysS cells in LB or minimal media supplemented with $^{15}NH_4Cl$ (and $ZnCl_2$ for PHD finger-containing constructs). Cultures were induced with 0.5 mM IPTG and grown for 18 h at 16 °C. After harvesting, the GST-tagged proteins were purified using glutathione agarose (Thermo Scientific) in 20 mM Tris pH 7.5, 150-500 mM NaCl, 5 mM $MgCl_2$ and 2 mM DTT. For experiments requiring GST-tagged constructs, the proteins were eluted in a buffer containing 20 mM Tris pH 7.5, 150-500 mM NaCl, 5 mM $MgCl_2$, 50 mM reduced glutathione and 2 mM DTT. For crystallography and biochemical assays, the GST tag was cleaved overnight at 4 °C with TEV or PreScission protease, and proteins were further purified by size exclusion chromatography (SEC). Protein fractions were checked for purity by SDS-PAGE and concentrated in a concentrator (Millipore).

## NMR titration experiments
NMR experiments were performed at 298 K on Varian 600 MHz and 900 MHZ spectrometers equipped with a cryogenic probe. Chemical shift perturbation experiments were carried out using 0.1–0.2 mM uniformly $^{15}N$-labeled wild type or mutant proteins in 25 mM Tris pH 7.5 buffer, supplemented with 150 mM NaCl, 5 mM DTT and 5-7% $D_2O$. $^1H,^{15}N$ heteronuclear single quantum coherence (HSQC) or transverse relaxation optimized spectroscopy (TROSY) spectra were recorded in the absence and presence of increasing concentrations of ligands. $K_d$ values were calculated by a nonlinear least-squares analysis in Kaleidagraph using the equation:

$$\Delta\delta = \Delta\delta_{max}\left(([L]+[P]+K_d) - \sqrt{([L]+[P]+K_d)^2 - 4[P][L]}\right)/2[P] \quad (1)$$

where [L] is concentration of the peptide, [P] is concentration of the protein, $\Delta\delta$ is the observed normalized chemical shift change and $\Delta\delta_{max}$ is the normalized chemical shift change at saturation, calculated as

$$\Delta\delta = \sqrt{(\Delta\delta H)^2 + (\Delta\delta N/5)^2} \quad (2)$$

where $\delta$ is the chemical shift in parts per million (ppm).

## Tryptophan fluorescence
Fluorescence spectra were collected at 25 °C on a Fluoromax-3 and Fluoromax-4 plus C spectrofluorometers (HORIBA). Samples contained 0.5–2 µM protein in a buffer containing 20–25 mM Tris pH 7.5, 150 mM NaCl and 3 mM DTT. Protein samples in the absence and presence of increasing concentrations of the histone peptides were excited at 295 nm. Emission spectra were recorded between 320 and 360 nm with a 0.5 nm step size and a 0.5 s integration time. The $K_d$ values were determined using a nonlinear least-squares analysis and the equation:

$$\Delta I = \Delta I_{max}\frac{(([L]+[P]+K_d) - \sqrt{([L]+[P]+K_d)^2 - 4[P][L]})}{2[P]} \quad (3)$$

where [L] is concentration of the peptide, [P] is concentration of the protein, $\Delta I$ is the observed change of signal intensity, and $\Delta I_{max}$ is the difference in signal intensity of the free and bound states of the protein. The $K_d$ values were averaged over three independent experiments with error calculated as the standard deviation between the runs.

## MST
Microscale thermophoresis (MST) experiments were performed using a Monolith NT.115 instrument (NanoTemper). Experiments were performed using SEC purified ASH1L$_{PHD}$ in buffer containing 50 mM Tris-HCl (pH 7.5) and 150 mM NaCl. The concentration of fluorophore, C-terminally 5-carboxyfluorescein (FAM)-labeled H3K4me3 peptide (Synpeptide, 1–12 aa), was 20 nM. Dissociation constant was determined using a direct binding assay in which increasing amounts of unlabeled ASH1L$_{PHD}$ protein was added stepwise. The measurements were performed at 90% LED and medium MST power with 3 s steady state, up to 20 s laser on time and 1 s off time. The $K_d$ value was calculated using MO Affinity Analysis software (NanoTemper) (n = 4). Figure was generated in GraphPad PRISM.

## X-ray crystallography
ASH1L$_{PHD}$ (12 mg/ml) was incubated with H3K4me2 or H3K4me3 (aa 1-12 of H3) peptide at a 1:1.5 molar ratio. Crystals of the ASH1L$_{PHD}$-H3K4me2/3 complexes were obtained using hanging-drop vapor diffusion method in 1.6 M sodium citrate pH 6.5 at 4 °C. ASH1L$_{PHD-BAH}$ (4 mg/ml) was incubated with H3K4me3 (aa 1-12 of H3) peptide at a 1:3 molar ratio. Crystals of ASH1L$_{PHD-BAH}$ were obtained using sitting-drop vapor diffusion method in 0.1 M Bis-Tris propane pH 10, 0.25 M strontium chloride, 0.01 M ammonium sulfate, and 25% PEG 8000 at 18 °C.

The datasets for the ASH1L$_{PHD}$-H3K4me2/3 complexes were collected at the National Synchrotron Light Source using beamline X25. The datasets were indexed, integrated and scaled using iMOSFLM and SCALA. The structure of ASH1L$_{PHD}$ in complex with H3K4me3 was solved by SAD using the data collected at peak wavelength of 1.28 Å. The structure of ASH1L$_{PHD}$ in complex with H3K4me2 was obtained by molecular replacement using the structure of ASH1L$_{PHD}$-H3K4me3 as a search model. Refinement of the models were carried out using Phenix_Refine and manually in Coot. The dataset for ASH1L$_{PHD-BAH}$ was collected at the University of Colorado Anschutz X-ray Core facility. The dataset was indexed, integrated and scaled

using XDS. MrBUMP was used to generate search models and determine the phases by molecular replacement. Initial model building and phase improvement was performed with BUCANEER. Refinement of the model was carried out using REFMAC and manually in Coot. All structures were validated by MolProbity and wwPDB OneDep System. Crystallographic statistics for the structures are shown in Supplementary Table 1.

### Protein structure determination by NMR
NMR spectra were collected at 298 K on Bruker 900 and 800 MHz NMR spectrometers equipped with z-gradient triple-resonance cryoprobes (Bruker Top-Spin v3.0). NMR samples contained 0.5 mM ASH1L$_{BD-PHD}$ (aa 2436-2638) and H3K4me2 peptide (aa 1-15 of H3) in a 200 mM sodium phosphate buffer (pH 6.3), supplemented with 5 mM perdeuterated dithiothreitol and H$_2$O/$^2$H$_2$O (9/1) or $^2$H$_2$O. The $^1$H, $^{13}$C, and $^{15}$N resonances of the protein in the complex were assigned using three-dimensional triple-resonance NMR experiments (HNCA, HN(CO)CA, HN(CA)CB and HN(COCA)CB) collected on a $^{13}$C/$^{15}$N-labeled and 75% deuterated protein bound to an unlabeled peptide. Distance restraints were obtained from three-dimensional $^{13}$C- and $^{15}$N-NOESY spectra. The H3K4me2 peptide resonances were assigned using 2D TOCSY, NOESY, and ROESY spectra. Intermolecular NOEs were obtained from $^{13}$C-edited (F$_1$), $^{13}$C/$^{15}$N-filtered (F$_3$) 3D NOESY spectra. NMR data were processed and analyzed by NMRPipe and NMRVIEW. Structures of the ASH1L$_{BD-PHD}$/H3K4me2 complex were determined using a distance geometry-simulated annealing protocol in CNS. Initial protein structure calculations were performed with manually assigned NOE-derived distance constraints. Hydrogen-bond distance restraints and φ and ψ dihedral-angle restraints from the TALOS+ prediction were added at a later stage of the structure calculations for the residues showing characteristic NOE patterns. The converged structures were used for the iterative automated NOE assignment by ARIA refinement. Structure quality was assessed with CNS, ARIA and PROCHECK analysis. A family of 200 structures was generated and 20 structures with the lowest energies were selected for the final analysis. Structure statistics are shown in Supplementary Table 2.

### Electrophoretic Mobility Shift Assay
The 147 bp 601 Widom DNA was generated by PCR. Briefly, the 147 bp DNA containing the 601 sequence in a pJ201 vector was amplified using Platinum Hot Start (Invitrogen) with primers 5′ ATCGAGAATCC CGGTGCCGAGGCCG and 3′ ATCGGATGTATATATCTGACACGTGC. Amplified 601 was then purified using Qiagen PCR purification kit. The 147 bp H3K4me3-NCP, unmodified 147bp-NCP and unmodified 187bp-NCP were purchased from Epicypher (Cat # 16-1315, 16-0009, and 16-2104). EMSAs were performed by mixing increasing amounts of purified ASH1L protein with 0.05 μM (final) 147 bp 601 Widom DNA or 0.025 μM (final) nucleosome in 25 mM Tris pH 7.5, 20 mM NaCl, 20% glycerol, 2 mM DTT, and 1 mM EDTA in a 10 μL reaction volume. Each sample was incubated on ice for 5 min and then loaded onto a 5% (59:1: acrylamide:bisacrylamide) native polyacrylamide gel. Electrophoresis of the gel was performed in 0.2x Tris-borate-EDTA (TBE) at 100 V for 90 min. The gels were stained with SYBR Gold (Thermo Fisher Sci) and visualized by Blue LED (UltraThin LED Illuminator-GelCompany). Each EMSA experiment was performed at least in duplicate.

### KMT assays
Purified wild-type and mutated GST-ASH1L$_{SET-BD-PHD}$ (1 μg) were incubated with nucleosomes (1 μg) in KMT reaction buffer (50 mM Tris-HCl, pH 8.0, 10% glycerol, 20 mM KCl, 5 mM MgCl$_2$, 1 mM DTT, 1 mM PMSF and 0.1 mM SAM) at 30 °C for 4 and 8 h. Reactions were quenched by flash-freezing in liquid nitrogen and then analyzed by SDS-PAGE and Western blot. Antibodies used: GST (Sata Cruz, Sc-459; 1:1000), H4 (Abcam, ab7311; 1:500), H3K4me3 (Abcam, ab8580; 1:1000), H3K36me1 (Abcam, ab9048; 1:1000), H3K36me2 (Abcam, ab9049; 1:5000).

### HMT assays
Recombinant ASH1L (WT and mutant) constructs, tagged at the N-terminus with GST and at the C-terminus with 6XHis, were generated and purified using Sepharose beads and Ni-NTA beads. Strep-MRG15 was purified using Baculovirus vector and infection of Sf9 cells following standard procedure. Methyltransferase activity of the proteins was measured with 0.5 μCi of $^3$H labeled S-adenosyl methionine (18 Ci/mmol; PerkinElmer Life Sciences). 20-100 ng of recombinant ASH1L, normalized by Coomassie and Western blotting, were used in the HMT assays. The reactions were performed in a volume of 30 μl using 0.5 μg oligonucleosomes from HeLa cells or 0.5 μg recombinant nucleosomes (NCP, Epicypher) as substrates in HMT buffer (20 mM Tris-HCl pH 8, 5% glycerol, 0.1 mM EDTA, 1 mM DTT, 1 mM PMSF) supplemented with 2 mM MgCl$_2$. Recombinant ASH1L were pre-incubated with the substrates and MRG15 (20 mM) or BSA for 30 min at 4 °C before addition of SAM, followed by 60 minutes at 30 °C. The reactions were captured on P81 filter paper, the free $^3$H-labeled SAM was washed away, and the paper was analyzed by Liquid Scintillation.

### Statistics and reproducibility
Tryptophan fluorescence data are presented as mean values of three independent measurements ± SD and MST data are presented as average of four independent measurements ± SEM. EMSA experiments in Figs. 4 and 5 and Supplementary Figs. 10 and 11 were performed at least twice.

### Reporting summary
Further information on research design is available in the Nature Portfolio Reporting Summary linked to this article.

## Data availability
Coordinates and structure factors have been deposited in the Protein Data Bank under the accession numbers 8VLD, 8VLF, 8VLH, and 8ZXC. NMR data have been deposited in the Biological Magnetic Resonance Bank under accession number 36675. The ChIP-seq data are deposited to GEO under accession number GSE199438. The RNA-seq data are deposited to GEO under accession number GSE198706. All other relevant data supporting the key findings of this study are available within the article and Supplementary Information files. Source data are provided with this paper.

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

## Acknowledgements

We thank Celine Roques for help with endogenous ASH1L experiments. This work was supported in part by grants from the NIH: CA252707, HL151334 and AG067664 to T.G.K., CA239165, AI124565 and AG072562 to M.-M.Z., CA204020 and CA268440 to X.S., R35GM152184 to S.B.R., GM126900 to B.D.S. and from CIHR (FND-143314) to J.C. NMR spectrometers at the Icahn School of Medicine at Mount Sinai supported with the grants from the NIH: OD025132 and OD028504.

## Author contributions

K.R.V., R.S., C-C.H., M.D., A.H.T., L.Z., K.L., L.Z., Q.L., C.L., R.R.O., Q.T., K.L.C., S.Y., S.B., H.X., J.G., L.A., J.W., S.M.J., B.J.K., Y.L., and S.B.R. performed experiments and together with P.E., B.D.S., M.J.W., M.L.C., J.C., X.S., M.-M.Z. and T.G.K. analyzed the data. K.R.V., R.S., M.-M.Z. and T.G.K. wrote the manuscript with input from all authors.

## Competing interests

The authors declare no competing interests.
