## [Transparent Peer Review file · Nature Communications]

Structure-function relationship of ASH1L and histone H3K36 and H3K4 methylation

Corresponding Author: Professor Tatiana Kutateladze

Version 1:

Reviewer comments:

Reviewer #1

(Remarks to the Author)

In this manuscript, Vann et al. report their investigation of human ASH1L, a large methyltransferase that specifically produces H3K36me₂. In addition to a catalytic SET domain, ASH1L encompasses a bromodomain (BD), a PHD domain, and a BAH domain with poorly understood functions. Using multiple complementary approaches, including X-ray crystallography, NMR spectroscopy, fluorescence spectroscopy, MST, and EMSA, the authors demonstrate that ASH1L-PHD binds H3K4me_{2/3}, while ASH1L-BD and ASH1L-BAH bind DNA. They show that the ASH1L PHD and BAH domains form a unique integrated module in which the PHD domain stabilizes the BAH fold. These extensive structural and biophysical characterizations are complemented by studies in cells, convincingly supporting a model where ASH1L generates H3K36me₂ at genomic sites with low H3K4me₃ levels. This study also connects ASH1L to cancer, showing that ASH1L depletion reduces the growth of lung adenocarcinoma cells. Furthermore, several mutations in the PHD domain are linked to cancer, with some mutations shown to diminish binding to an H3K4me₃ peptide. Given the previously reported anti-leukemic activity of ASH1L inhibition, the study by Vann et al. has the potential to facilitate the development of therapeutic small molecule inhibitors. The manuscript is clear and well-written, and the work is of excellent technical quality. I enthusiastically support its publication.

Reviewer #2

(Remarks to the Author)

This manuscript investigates the ASH1L protein, focusing on its structural and functional roles in histone H3K36 and H3K4 methylation. The study employs a combination of structural biology techniques, including NMR spectroscopy and X-ray crystallography, as well as functional assays like RNA-seq and ChIP-seq, to elucidate the molecular mechanisms by which ASH1L interacts with chromatin and regulates gene expression. The findings are significant for understanding the role of ASH1L in embryonic stem cell differentiation and its broader implications in disease. However, there are the following issues that need to be addressed.

1. Consider revising the abstract to more explicitly highlight the novelty and potential impact of the study.
2. Include a more detailed explanation of the statistical methods employed for RNA-seq and ChIP-seq data analysis. Clearly define the controls used in each experiment to enhance the methodological transparency.
3. Improve the clarity of some figures, particularly the binding affinity graphs, to ensure they are easily interpretable by readers.
4. Authors should provide the original uncropped minimally adjusted images supporting at least three independent replicate assays blot and gel results in an article's figures and supporting information files.
5. Include a discussion of the study's limitations and offer a more detailed outline of potential future research directions.
6. Expand on the broader implications of the research, particularly in terms of its potential impact on therapeutic approaches and disease understanding.

In general, the manuscript is well-structured and contributes valuable insights into the role of ASH1L in chromatin regulation. While the study is methodologically sound and the results are significant, minor to moderate revisions are necessary to enhance the clarity and depth of the manuscript. Specifically, the authors should focus on improving figure clarity, providing more detailed discussions of statistical methods and experimental controls, addressing the study's limitations, and expanding on the broader implications of their findings.

Reviewer #3

(Remarks to the Author)

The manuscript by Kutateladze and coworkers investigates the structure and function of the C-terminal bromodomain (BD),

plant homeodomain (PHD) and bromo-associated homology (BAH) domain in the histone H3 Lys36 (H3K36) methyltransferase ASH1L. Overexpression and mutations of ASH1L are associated with autoimmune disease, cancer, and neurological disorders. To gain insights into its biological functions and dysregulation in disease, the authors analyzed the expression of ASH1L during embryonic stem (ES) cell differentiation and found that the expression of the enzyme is upregulated in different cell lineages. Further, knockout or knockdown of the ASH1L gene disrupted stem cell differentiation, illustrating a role for ASH1L in regulating cellular differentiation. ChIP-Seq analysis demonstrated ASH1L localized to transcription start sites (TSSs) of genes that were marked by H3K4 trimethylation (H3K4me3) but not at sites enriched in H3K36me2, the modification catalyzed by the enzyme. To investigate the inverse correlation between the enzyme's chromatin localization and its methyltransferase activity, the authors characterized the ASH1L PHD, BD-PHD, PHD-BAH, and BD-PHD-BAH domains using a combination of structural and biochemical approaches. Their studies revealed that the PHD domain recognizes H3K4me2 and H3K4me3, whereas the BD and BAH domains possess positively charged surfaces that mediate DNA binding. In addition, the PHD-BAH tandem domains were shown to bind to nucleosome core particles (NCPs). Complementing the binding assays, the ASH1L-SET-BD-PHD-BAH protein was shown to methylate NCPs, whereas its activity was diminished toward H3K4me3 NCPs. These findings illustrate that the binding of H3K4me3 to the PHD domain inhibits the catalytic activity of the SET domain of ASH1L, offering a potential explanation for the observed inverse correlation between H3K36me2 and ASH1L's localization to H3K4me3-enriched TSSs. Collectively, the authors' work provides novel insights into the functions of the BD, PHD, and BAH domains in chromatin engagement and in regulating the methyltransferase activity of ASH1L. These studies merit publication pending the following points are addressed.

1. The authors have shown that the ASH1L PHD-BAH tandem domains bind to NCPs. Given that ASH1L BD can recognize DNA, it would be worthwhile investigating the binding of the ASH1L BD-PHD and BD-PHD-BAH tandem domains to NCPs to gain a more complete understanding of how chromatin engagement occurs with the three domains.

2. The interactions between the ASH1L BD and H3K56ac were mapped to a region outside the canonical acetyllysine binding pocket (Supplementary Figure 9). This is an interesting observation, and a brief description and/or figure of the H3K56ac binding site and its amino acid composition would be enlightening. Also, does the H3K56ac binding site in the ASH1L BD lie within or outside the positively charged surfaces that mediate DNA binding?

3. The effect of the ASH1L PHD D2595K mutation in abolishing binding to H3K4me3 is attributed to electrostatic repulsion between K2595 and K4me3. However, D2595 forms a hydrogen bond with the side chain of H3T6, and it is conceivable that the D2595K mutation also disrupts this hydrogen bond and may result in a steric clash with T6. This should be noted in the manuscript. Testing a D2595A mutant would abolish electrostatic interactions with K4me3 while circumventing any potential steric clash with T6.

4. Related to #3, D2584 also lies within the K4me3 binding site in the ASH1L PHD domain and is positioned to form favorable electrostatic interactions with the K4me3 side chain. It would be useful to examine whether a D2584 mutation affects H3K4me3 binding by the PHD domain, similar to mutation of D2595.

Version 2:

Reviewer comments:

Reviewer #2

(Remarks to the Author)

I would like to thank the authors for their diligent work in addressing the concerns raised in my previous review. After thoroughly evaluating the revised manuscript, I am pleased to note that the authors have effectively resolved the issues discussed. I believe the revisions have significantly enhanced the clarity and quality of the manuscript. Therefore, I recommend that the paper be accepted for publication.

Reviewer #3

(Remarks to the Author)

The revisions by Kutateladze and coworkers have addressed the comments and strengthened their manuscript. This work is recommended for publication.

We would like to thank the Reviewers for their insightful and very constructive comments, which were very helpful in revising and strengthening this manuscript.

Reviewer #1 (Remarks to the Author):

In this manuscript, Vann et al. report their investigation of human ASH1L, a large methyltransferase that specifically produces H3K36me2. In addition to a catalytic SET domain, ASH1L encompasses a bromodomain (BD), a PHD domain, and a BAH domain with poorly understood functions. Using multiple complementary approaches, including X-ray crystallography, NMR spectroscopy, fluorescence spectroscopy, MST, and EMSA, the authors demonstrate that ASH1L-PHD binds H3K4me2/3, while ASH1L-BD and ASH1L-BAH bind DNA. They show that the ASH1L PHD and BAH domains form a unique integrated module in which the PHD domain stabilizes the BAH fold. These extensive structural and biophysical characterizations are complemented by studies in cells, convincingly supporting a model where ASH1L generates H3K36me2 at genomic sites with low H3K4me3 levels. This study also connects ASH1L to cancer, showing that ASH1L depletion reduces the growth of lung adenocarcinoma cells. Furthermore, several mutations in the PHD domain are linked to cancer, with some mutations shown to diminish binding to an H3K4me3 peptide. Given the previously reported anti-leukemic activity of ASH1L inhibition, the study by Vann et al. has the potential to facilitate the development of therapeutic small molecule inhibitors. The manuscript is clear and well-written, and the work is of excellent technical quality. I enthusiastically support its publication.

Reviewer #2 (Remarks to the Author):

This manuscript investigates the ASH1L protein, focusing on its structural and functional roles in histone H3K36 and H3K4 methylation. The study employs a combination of structural biology techniques, including NMR spectroscopy and X-ray crystallography, as well as functional assays like RNA-seq and ChIP-seq, to elucidate the molecular mechanisms by which ASH1L interacts with chromatin and regulates gene expression. The findings are significant for understanding the role of ASH1L in embryonic stem cell differentiation and its broader implications in disease. However, there are the following issues that need to be addressed.

1. Consider revising the abstract to more explicitly highlight the novelty and potential impact of the study. – we have revised abstract as suggested.
2. Include a more detailed explanation of the statistical methods employed for RNA-seq and ChIP-seq data analysis. Clearly define the controls used in each experiment to enhance the methodological transparency. – the RNA-seq and ChIP-seq method sections have been revised to include more details and Suppl. Fig. 1 (volcano plot) has been added.
3. Improve the clarity of some figures, particularly the binding affinity graphs, to ensure they are easily interpretable by readers. – we have updated Fig. 1g with error bars and statistical analysis and clarified normalization equation for the binding graphs in Fig. 3e legend.
4. Authors should provide the original uncropped minimally adjusted images supporting at least three independent replicate assays blot and gel results in an article's figures and supporting information files. – uncropped gels and other raw data will be included in the Data Source file.
5. Include a discussion of the study's limitations and offer a more detailed outline of potential future research directions. – we have revised the first paragraph of discussion to include this (page 14).
6. Expand on the broader implications of the research, particularly in terms of its potential impact on therapeutic approaches and disease understanding. – this is now highlighted in the second paragraph of discussion (page 15).

In general, the manuscript is well-structured and contributes valuable insights into the role of ASH1L in chromatin regulation. While the study is methodologically sound and the results are significant, minor to moderate revisions are necessary to enhance the clarity and depth of the manuscript. Specifically, the

authors should focus on improving figure clarity, providing more detailed discussions of statistical methods and experimental controls, addressing the study's limitations, and expanding on the broader implications of their findings.

Reviewer #3 (Remarks to the Author):

The manuscript by Kutateladze and coworkers investigates the structure and function of the C-terminal bromodomain (BD), plant homeodomain (PHD) and bromo-associated homology (BAH) domain in the histone H3 Lys36 (H3K36) methyltransferase ASH1L. Overexpression and mutations of ASH1L are associated with autoimmune disease, cancer, and neurological disorders. To gain insights into its biological functions and dysregulation in disease, the authors analyzed the expression of ASH1L during embryonic stem (ES) cell differentiation and found that the expression of the enzyme is upregulated in different cell lineages. Further, knockout or knockdown of the ASH1L gene disrupted stem cell differentiation, illustrating a role for ASH1L in regulating cellular differentiation. ChIP-Seq analysis demonstrated ASH1L localized to transcription start sites (TSSs) of genes that were marked by H3K4 trimethylation (H3K4me3) but not at sites enriched in H3K36me2, the modification catalyzed by the enzyme. To investigate the inverse correlation between the enzyme's chromatin localization and its methyltransferase activity, the authors characterized the ASH1L PHD, BD-PHD, PHD-BAH, and BD-PHD-BAH domains using a combination of structural and biochemical approaches. Their studies revealed that the PHD domain recognizes H3K4me2 and H3K4me3, whereas the BD and BAH domains possess positively charged surfaces that mediate DNA binding. In addition, the PHD-BAH tandem domains were shown to bind to nucleosome core particles (NCPs). Complementing the binding assays, the ASH1L-SET-BD-PHD-BAH protein was shown to methylate NCPs, whereas its activity was diminished toward H3K4me3 NCPs. These findings illustrate that the binding of H3K4me3 to the PHD domain inhibits the catalytic activity of the SET domain of ASH1L, offering a potential explanation for the observed inverse correlation between H3K36me2 and ASH1L's localization to H3K4me3-enriched TSSs. Collectively, the authors' work provides novel insights into the functions of the BD, PHD, and BAH domains in chromatin engagement and in regulating the methyltransferase activity of ASH1L. These studies merit publication pending the following points are addressed.

1. The authors have shown that the ASH1L PHD-BAH tandem domains bind to NCPs. Given that ASH1L BD can recognize DNA, it would be worthwhile investigating the binding of the ASH1L BD-PHD and BD-PHD-BAH tandem domains to NCPs to gain a more complete understanding of how chromatin engagement occurs with the three domains.

Author's response: as suggested, we have performed EMSA assays of 187bp-, 147bp- and H3K4me3-NCPs and 601 DNA using BD-PHD-BAH, BD-PHD and PHD (Suppl. Fig. 11). The data indicate that binding of BD-PHD-BAH to NCPs is not appreciably affected by BD, but the preference for 187bp-NCP and H3K4me3-NCP over 147bp-NCP remains for either PHD-BAH or BD-PHD-BAH. PHD itself is incapable of binding to H3K4me3-NCP, in support of previous studies reporting that histone tails are not easily accessible in context of the nucleosome and have to be released from the nucleosomal DNA to be bound by readers. The presence of PHD in BD-PHD decreased binding to DNA, likely due to the negatively charged linker between the two domains as well as the negatively charged surface of PHD, however the binding to DNA was restored for the BD-PHD-BAH construct.

2. The interactions between the ASH1L BD and H3K56ac were mapped to a region outside the canonical acetyllysine binding pocket (Supplementary Figure 9). This is an interesting observation, and a brief description and/or figure of the H3K56ac binding site and its amino acid composition would be

enlightening. Also, does the H3K56ac binding site in the ASH1L BD lie within or outside the positively charged surfaces that mediate DNA binding? – we have mapped the H3K56ac-binding and DNA-binding residues in Suppl. Fig. 9 and added the following text on page 10: ‘...revealed two highly positively charged surface regions located far from the H3K56ac binding site (Suppl. Fig. 9).’

3. The effect of the ASH1L PHD D2595K mutation in abolishing binding to H3K4me3 is attributed to electrostatic repulsion between K2595 and K4me3. However, D2595 forms a hydrogen bond with the side chain of H3T6, and it is conceivable that the D2595K mutation also disrupts this hydrogen bond and may result in a steric clash with T6. This should be noted in the manuscript. Testing a D2595A mutant would abolish electrostatic interactions with K4me3 while circumventing any potential steric clash with T6.

4. Related to #3, D2584 also lies within the K4me3 binding site in the ASH1L PHD domain and is positioned to form favorable electrostatic interactions with the K4me3 side chain. It would be useful to examine whether a D2584 mutation affects H3K4me3 binding by the PHD domain, similar to mutation of D2595. – as suggested, we have generated D2595A and D2584 mutants and measured their binding affinities to H3K4me3 (the data are shown in Fig. 3c and Suppl. Fig. 5). The following text has been added on page 9: ‘Substitution of D2595 or D2584 with alanine decreased binding affinity to ~5-7 μ M, indicating the importance of the hydrogen bond of D2595 with T6 and favorable electrostatic contacts of D2595 and D2584 with K4me3 for the complex formation (Fig. 3c and Suppl. Fig. 5).’